# TIGHT NON-ASYMPTOTIC INFERENCE VIA SUB-GAUSSIAN INTRINSIC MOMENT NORM

## ABSTRACT

In non-asymptotic statistical inferences, variance-type parameters of sub-Gaussian distributions play a crucial role. However, direct estimation of these parameters based on the empirical moment generating function (MGF) is infeasible. To this end, we recommend using a sub-Gaussian intrinsic moment norm [Buldygin and Kozachenko (2000), Theorem 1.3] through maximizing a series of normalized moments. Importantly, the recommended norm can not only recover the exponential moment bounds for the corresponding MGFs, but also lead to tighter Hoeffding's sub-Gaussian concentration inequalities. In practice, we propose an intuitive way of checking sub-Gaussian data with a finite sample size by the sub-Gaussian plot. Intrinsic moment norm can be robustly estimated via a simple plug-in approach. Our theoretical results are applied to non-asymptotic analysis, including the multi-armed bandit.

## 1 INTRODUCTION

With the advancement of machine learning techniques, computer scientists have become more interested in establishing rigorous error bounds for desired learning procedures, especially those with finite sample validity (Wainwright, 2019; Zhang & Chen, 2021; Yang et al., 2020). In specific settings, statisticians, econometricians, engineers and physicist have developed non-asymptotic inferences to quantify uncertainty in data; see Romano & Wolf (2000); Chassang (2009); Arlot et al. (2010); Yang et al. (2020); Horowitz & Lee (2020); Armstrong & Kolesár (2021); Zheng & Cheng (2021); Lucas et al. (2008); Owhadi et al. (2013); Wang (2020). Therefore, the concentration-based statistical inference has received a considerable amount of attention, especially for bounded data (Romano & Wolf, 2000; Auer et al., 2002; Hao et al., 2019; Wang et al., 2021; Shiu, 2022) and Gaussian data (Arlot et al., 2010; Duy & Takeuchi, 2022; Bettache et al., 2021; Feng et al., 2021). For example, Hoeffding's inequality can be applied to construct non-asymptotic confidence intervals based on bounded data[1].

However, in reality, it may be hard to know the support of data or its underlying distribution. In this case, misusing Hoeffding's inequality (Hoeffding, 1963) for unbounded data will result in a notably loose confidence interval (CI); see Appendix A.1. Hence, it is a common practice to assume that data follow sub-Gaussian distribution (Kahane, 1960). By the Chernoff inequality[2], we have $P(X \geq t) \leq \inf_{s>0}\{\exp\{-st\}E\exp\{sX\}\}$, $\forall t \geq 0$. Hence, tightness of a confidence interval relies on how we upper bound the moment generating function (MGF) $E\exp\{sX\}$ for all $s > 0$. This can be further translated into the following optimal variance proxy of sub-Gaussian distribution.

**Definition 1.** *A r.v. $X$ is sub-Gaussian (sub-G) with a variance proxy $\sigma^2$ [denoted as $X \sim \mathrm{subG}(\sigma^2)$] if its MGF satisfies $E\exp(tX) \leq \exp(\sigma^2 t^2/2)$ for all $t \in \mathbb{R}$. The sub-Gaussian parameter $\sigma_{opt}(X)$ is defined by the optimal variance proxy (Chow, 1966):*

$$\sigma_{opt}^2(X) := \inf\left\{\sigma^2 > 0 : E\exp(tX) \leq \exp\{\sigma^2 t^2/2\}, \quad \forall t \in \mathbb{R}\right\} = 2\sup_{t \in \mathbb{R}} t^{-2}\log[E\exp(tX)]. \quad (1)$$

Note that $\sigma_{opt}^2(X) \geq \mathrm{Var}\,X$; see (14) in Appendix A.2. When $\sigma_{opt}^2(X) = \mathrm{Var}\,X$, it is called strict sub-Gaussianity (Arbel et al., 2020). Based on Theorems 1.5 in Buldygin & Kozachenko (2000), we have

$$P(X \geq t) \leq \exp\left\{-\frac{t^2}{2\sigma_{opt}^2(X)}\right\}, \quad P\left(|\sum_{i=1}^{n} X_i| \geq t\right) \leq 2\exp\left\{-\frac{t^2}{2\sum_{i=1}^{n} \sigma_{opt}^2(X_i)}\right\}. \quad (2)$$

---

[1]Recently, Phan et al. (2021) obtained a sharper result than Hoeffding's inequality for bounded data.
[2]For simplicity, we consider centered random variable (r.v.) with zero mean throughout the paper for all sub-Gaussian r.v..

for independent sub-G r.v.s $X$ and $\{X_i\}_{i=1}^n$. The above inequality (2) provides the tightest upper bound over the form $P(X > t) \leq \exp(-Ct^2)$ (or $P(|\sum_{i=1}^n X_i| > t) \leq \exp(-Ct^2)$) for some positive constant $C$ via Chernoff inequality.

Given $\{X_i\}_{i=1}^n \overset{\text{i.i.d.}}{\sim} \mathrm{subG}(\sigma_{opt}^2(X))$, a straightforward application of (2) gives an non-asymptotic $100(1-\alpha)\%$ CI

$$\mathrm{E}X = 0 \in [\overline{X}_n \pm \sigma_{opt}(X)\sqrt{2n^{-1}\log(2/\alpha)}]. \tag{3}$$

A naive plug-in estimate[3] of $\sigma_{opt}^2(X) := 2\sup_{t\in\mathbb{R}} t^{-2}\log[\mathrm{E}\exp(tX)]$ (Arbel et al., 2020) is

$$\widehat{\sigma}_{opt}^2(X) := 2\sup_{t\in\mathbb{R}} t^{-2}\log[n^{-1}\Sigma_{i=1}^n \exp(tX_i)]. \tag{4}$$

However, two weaknesses of (4) substantially hinder its application: (i) the optimization result is unstable due to the possible non-convexity of the objective function; (ii) exponentially large $n$ is required to ensure the variance term $\mathrm{Var}(n^{-1}\sum_{i=1}^n \exp(tX_i))$ not to explode when $t$ is large. In Section 3, we present some simulation evidence.

On the other hand, we are aware of other forms of variance-type parameter. For instance, van der Vaart & Wellner (1996) introduced the Orlicz norm as $\|X\|_{w_2} := \inf\{c > 0 : \mathrm{E}\exp\{|X|^2/c^2\} \leq 2\}$, frequently used in empirical process theory. Additionally, Vershynin (2010) suggested a norm based on the scale of moments as $\|X\|_{\psi_2} := \max_{k\geq 2} k^{-1/2}(\mathrm{E}|X|^k)^{1/k}$ in Page 6 of Buldygin & Kozachenko (2000). However, as shown in Table 1 and Appendix A.2.1, both types of norm fail to deliver sharp probability bounds even for strict sub-G distributions, such as the standard Gaussian distribution and symmetric beta distribution.

Table 1: Comparison of sub-Gaussian norms $\|\cdot\|_*$ for centralized and symmetric $X$.

| $\|\cdot\|_*$-norm | sharp tail for $P(|X| \geq t)$ | sharp MGF bound | half length of $(1-\delta)$-CI | easy to estimate |
|---|---|---|---|---|
| $\sigma_{opt}(X)$ | Yes $[2\exp\{-\frac{t^2}{2}/\sigma_{opt}^2(X)\}]$ | Yes $[\exp\{\sigma_{opt}^2(X)\frac{t^2}{2}\}]$ | $\sqrt{2\log(2/\delta)}\sigma_{opt}(X)$ | No |
| $\|X\|_{w_2}$ | Yes $[2\exp\{-\frac{t^2}{2}/(\frac{\|X\|_{w_2}}{\sqrt{2}})^2\}]$ | No $[\exp\{(2\|X\|_{w_2})^2\frac{t^2}{2}\}]$ | $\sqrt{2\log(2/\delta)}\|X\|_{w_2}/\sqrt{2}$ | No |
| $\|X\|_{\psi_2}$ | No $[2\exp\{-\frac{t^2}{2}/(2e\|X\|_{\psi_2}^2)\}]$ | No $[\exp\{(4\sqrt{e}\|X\|_{\psi_2})^2\frac{t^2}{2}\}]$ | $\sqrt{2\log(2/\delta)}\sqrt{2e}\|X\|_{\psi_2}$ | Yes |
| $\|X\|_G$ (Def. 2) | Yes $[2\exp\{-\frac{t^2}{2}/\|X\|_G^2\}]$ | Yes $[\exp\{\|X\|_G^2\frac{t^2}{2}\}]$ | $\sqrt{2\log(2/\delta)}\|X\|_G$ | Yes |

## 1.1 CONTRIBUTIONS

In light of the above discussions, we advocate the use of the intrinsic moment norm in the Definition 2 in the construction of tight non-asymptotic CIs. There are two specific reasons: (i) it approximately recovers tight inequalities (2); (ii) it can be estimated friendly (with a closed form) and robustly.

The following definition 2 is from Page 6 and Theorem 1.3 in Buldygin & Kozachenko (2000).

**Definition 2** (Intrinsic moment norm). $\|X\|_G := \max_{k\geq 1} \left[\frac{2^k k!}{(2k)!}\mathrm{E}X^{2k}\right]^{1/(2k)} = \max_{k\geq 1}\left[\frac{1}{(2k-1)!!}\mathrm{E}X^{2k}\right]^{1/(2k)}$.

From the sub-G characterization (see Theorem 2.6 in Wainwright (2019)), $\|X\|_G < \infty$ iff $\sigma_{opt}(X) < \infty$ *for any zero-mean r.v.* $X$. Hence, the finite intrinsic moment norm of a r.v. $X$ ensures sub-Gaussianity (satisfying Definition 1).

Our contributions in this paper can be summarized as follows.

1. By $\|X\|_G$, we achieve a sharper Hoeffding-type inequality under asymetric distribution; see Theorem 2(b).

2. Compared to the normal approximation based on Berry-Esseen (B-E) bounds, our results are more applicable to data of extremely small sample size. We illustrate Bernoulli observations with the comparison of two types of CIs based on the B-E-corrected CLT and Hoeffding's inequality in Figure 1; see Appendix A for details.

3. A novel method called *sub-Gaussian plot* is proposed for checking whether the unbounded data are sub-Gaussian. We introduce plug-in and robust plug-in estimators for $\|X\|_G$, and establish finite sample theories.

4. Finally, we employ the intrinsic moment norm estimation in the non-asymptotic inference for a bandit problem: Bootstrapped UCB-algorithm for multi-armed bandits. This algorithm is shown to achieve feasible error bounds and competitive cumulative regret on unbounded sub-Gaussian data.

---

[3]We point out that a conservative and inconsistent estimator $2\inf_{t\in\mathbb{R}}\log(n^{-1}\sum_{i=1}^n \exp(tX_i))/t^2$ was proposed in statistical physics literature (Wang, 2020).

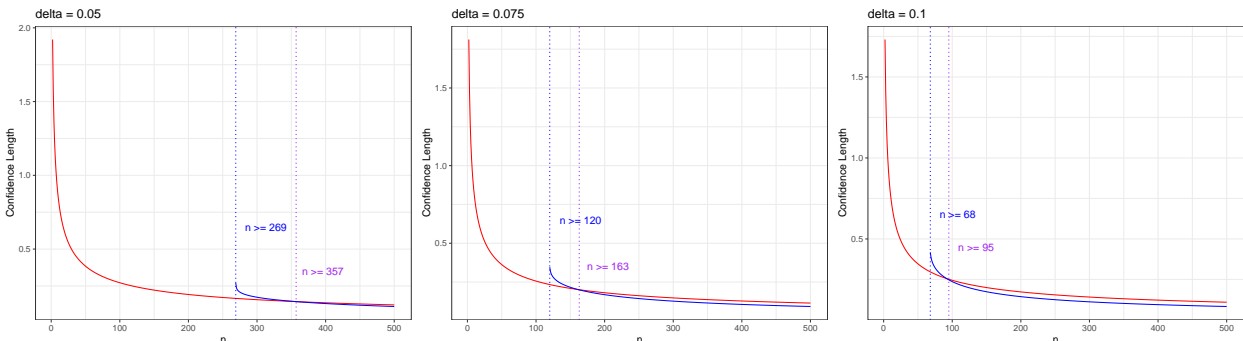

Figure 1: CIs via Hoeffding's inequality (red line) and B-E-corrected CLT (blue line). It describes a deficiency of B-E-corrected CLT under small sample, and it suggests that a simple Hoeffding's inequality can even perform better.

## 2   SUB-GAUSSIAN PLOT AND TESTING

Before estimating $\|X\|_G$, the first step is to verify $X$ is indeed sub-G given its i.i.d. copies $\{X_i\}_{i=1}^n$. Corollary 7.2 (b) in Zhang & Chen (2021) shows for r.v.s $X_i \sim \text{subG}(\sigma_{opt}^2(X))$ (without independence assumption)

$$\text{P}(\max_{1 \le i \le j} X_i \le \sigma_{opt}(X)\sqrt{2(\log j + t)}) \ge 1 - \exp\{-t\}, \tag{5}$$

which implies $\max_{1 \le i \le j} X_i = O_{\text{P}}(\sqrt{\log j})$. Moreover, we will show the above rate is indeed sharp for a class of unbound sub-G r.v.s characterized by the *lower intrinsic moment norm* below.

**Definition 3** (Lower intrinsic moment norm). *The lower intrinsic moment norm for a sub-G $X$ is defined as*

$$\|X\|_{\tilde{G}} := \min_{k \ge 1}\{[(2k-1)!!]^{-1}\text{E}X^{2k}\}^{1/(2k)}.$$

By the method in Theorem 1 of Zhang & Zhou (2020), we obtain the following tight rate result with a lower bound.

**Theorem 1.** *(a). If $\|X\|_{\tilde{G}} > 0$ for i.i.d. symmetric sub-G r.v.s $\{X_i\}_{i=1}^n \sim X$, then with probability at least $1 - \delta$*

$$\frac{\|X\|_{\tilde{G}}/\|X\|_G}{2\sqrt{2\|X\|_G^2/\|X\|_{\tilde{G}}^2 - 1}}\sqrt{\log n - \log C^{-2}(X) - \log\log\left(\frac{2}{\delta}\right)} \le \max_{1 \le i \le n}\frac{X_i}{\|X\|_G} \le \sqrt{2[\log n + \log\left(\frac{2}{\delta}\right)]},$$

*where $C(X) < 1$ is constant defined in Lemma 1 below; (b) if $X$ is bounded variable, then $\|X\|_{\tilde{G}} = 0$.*

The upper bound follows from the proof of (5) similarly. The proof of lower bound relies on the sharp reverse Chernoff inequality from Paley–Zygmund inequality (see Paley & Zygmund (1932)).

**Lemma 1** (A reverse Chernoff inequality). *Suppose $\|X\|_{\tilde{G}} > 0$ for a symmetric sub-G r.v. $X$. For $t > 0$, then*

$$\text{P}(X \ge t) \ge C^2(X)\exp\{-4[2\|X\|_G^2/\|X\|_{\tilde{G}}^4 - \|X\|_{\tilde{G}}^{-2}]t^2\},$$

*where $C(X) := \left(\frac{\|X\|_{\tilde{G}}^2}{4\|X\|_G^2 - \|X\|_{\tilde{G}}^2}\right)\left(\frac{4\|X\|_G^2 - 2\|X\|_{\tilde{G}}^2}{4\|X\|_G^2 - \|X\|_{\tilde{G}}^2}\right)^{2[2\|X\|_G^2/\|X\|_{\tilde{G}}^2 - 1]} \in (0, 1).$*

Theorem 1 of Zhang & Zhou (2020) does not optimize the constant in Paley–Zygmund inequality. In contrast, our Lemma 1 has an optimal constant; see Appendix C for details.

**Sub-Gaussian plot under unbounded assumption**[4]. By Theorem 1, we propose a novel *sub-Gaussian plot* check whether i.i.d data $\{X_i\}_{i=1}^n$ follow a sub-G distribution. Suppose that for each $j$, $\{X_i^*\}_{i=1}^j$ are independently sampled from the empirical distribution $\mathbb{F}_n(x) = \frac{1}{n}\sum_{i=1}^n 1(X_i \le x)$ of $\{X_i\}_{i=1}^n$. Specifically, we plot the order statistics $\{\max_{1 \le i \le j} X_i^*\}_{j=1}^n$ on the plane coordinate axis, where $x$ axis represents $\sqrt{\log j + 1}$ and $y$ axis the value of

---

[4]Sub-G plot can only be applied to data with enough samples. When $n$ is very small, there is not enough information to suggest unbounded trends. We roughly treat the data as bounded r.v. for a very small $n$, and there is no need to use a sub-G plot in this case.

$\max_{1 \leq i \leq j} X_i^*$. We check whether those points have a linear tendency at the boundary: the more they are close to the tendency of a beeline, the more we can trust the data are sub-Gaussian.

The Figure 2 shows *sub-Gaussian plot* of $N(0,1)$ and $\mathrm{Exp}(1)$. It can be seen that sub-Gaussian plot of $N(0,1)$ shows linear tendency at the boundary, while $\mathrm{Exp}(1)$ shows quadratic tendency at the boundary. For the quadratic tendency, we note that if $\{X_i\}_{i=1}^n$ have heavier tails such as sub-exponentiality, then $\max_{1 \leq i \leq j} X_i = O_P(\log j)$ instead of the order $O(\sqrt{\log j})$; see Corollary 7.3 in Zhang & Chen (2021).

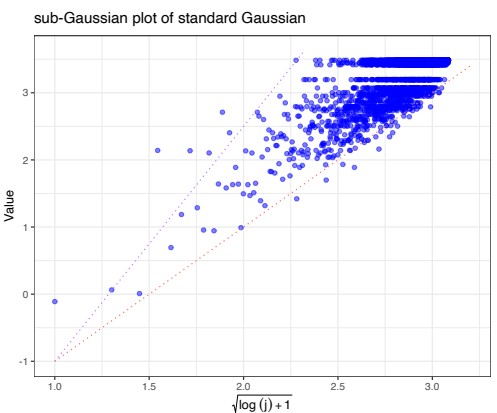
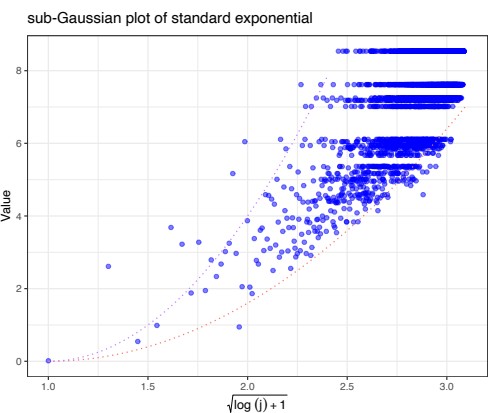

Figure 2: sub-Gaussian plot of standard Gaussian and standard exponential distribution for $n = 1000$. Left: The two dot lines indicate the points drop in a triangle region with a high probability. Right: The points in the case of exponential distribution approximately live curve triangle region with quadratic trends.

## 3 FINITE SAMPLE PROPERTIES OF INTRINSIC MOMENT NORM

In this section, we characterize two important properties of the intrinsic moment norm that are used in constructing non-asymptotic confidence intervals.

### 3.1 BASIC PROPERTIES

Lemma 2 below establishes that the intrinsic moment norm is estimable.

**Lemma 2.** *For sub-G $X$, we have* $\arg\max_{m \in 2\mathbb{N}} \left[ \frac{\mathrm{E}X^m}{(m-1)!!} \right]^{1/m} < \infty$, *where* $2\mathbb{N} := \{2, 4, \cdots\}$ *is the even number set.*

Lemma 2 ensure that for any sub-Gaussian variable $X$, its intrinsic moment norm can be computed as

$$\|X\|_G := \max_{m \in 2\mathbb{N}} \left[ \frac{\mathrm{E}X^m}{(m-1)!!} \right]^{1/m} = \max_{1 \leq k \leq k_X} \left[ \frac{\mathrm{E}X^{2k}}{(2k-1)!!} \right]^{1/(2k)} \text{ with some finite } k_X < \infty.$$

This is an important property that other norms may not have. The $\|X\|_{\psi_2} := \max_{k \geq 2} k^{-1/2} (\mathrm{E}|X|^k)^{1/k}$ for Gaussian $X$ achieves its optimal point at $k = \infty$; see Example 3 in Appendix A.2.1. As for $\sigma_{opt}^2(X) := 2 \sup_{t \in \mathbb{R}} \frac{\log[\mathrm{E}\exp(tX)]}{t^2}$, it is unclear that its value can be achieved at a finite $t$. Note that if $k_X = 1$, one has $\|X\|_G^2 = \mathrm{Var}(X)$.

Next, we present an example in calculating the values of $k_X$. Denote $\mathrm{Exp}(1)|_{[0,M]}$ as the truncated standard exponential distribution on $[0,M]$ with the density as $f(x) = \frac{e^{-x}}{\int_0^M e^{-x}\,dx} 1_{\{x \in [0,M]\}}$.

**Example 1.** *a.* $X \sim U[-a,a]$, $k_X = 1$ *for any* $a \in \mathbb{R}$; *b.* $X \sim \mathrm{Exp}(1)|_{[0,2.75]} - \mathrm{E}\,\mathrm{Exp}(1)|_{[0,2.75]}$, $k_X = 2$; *c.* $X \sim \mathrm{Exp}(1)|_{[0,3]} - \mathrm{E}\,\mathrm{Exp}(1)|_{[0,3]}$, $k_X = 3$. *Indeed, for any fixed* $k_0 \in \mathbb{N}$, *we can construct a truncated exponential r.v.* $X := \mathrm{Exp}(1)|_{[0,M]}$ *such that* $k_X = k_0$ *by properly adjusting the truncation level* $M$.

### 3.2 CONCENTRATION FOR SUMMATION

In what follows, we will show another property of $\|X\|_G$ that it recovers nearly tight MGF bounds in Definition 1. More powerfully, it enables us to derive the sub-G Hoeffding's inequality (2).

**Theorem 2.** *Suppose that $\{X_i\}_{i=1}^n$ are independent r.v.s with $\max_{i\in[n]}\|X_i\|_G < \infty$. We have*

(a). *If $X_i$ is symmetric about zero, then $\mathrm{E}\exp\{tX_i\} \le \exp\{t^2\|X_i\|_G^2/2\}$ for any $t\in\mathbb{R}$, and*

$$\mathrm{P}\left(|\sum_{i=1}^n X_i| \ge s\right) \le 2\exp\{-s^2/[2\sum_{i=1}^n\|X_i\|_G^2]\}, \quad s \ge 0.$$

(b). *If $X_i$ is not symmetric, then $\mathrm{E}\exp\{tX_i\} \le \exp\{(17/12)t^2\|X_i\|_G^2/2\}$ for any $t\in\mathbb{R}$, and*

$$\mathrm{P}\left(|\sum_{i=1}^n X_i| \ge s\right) \le 2\exp\{-(12/17)s^2/[2\sum_{i=1}^n\|X_i\|_G^2]\}, \quad s \ge 0.$$

Theorem 2(a) is an existing result in Theorem 2.6 of Wainwright (2019). For Theorem 2(b), we obtain $\sqrt{17/12} \approx 1.19$, while Lemma 1.5 in Buldygin & Kozachenko (2000) obtained $\mathrm{E}\exp\{tX_i\} \le \exp\left\{\frac{t^2}{2}(\sqrt[4]{3.1}\|X_i\|_G)^2\right\}$ for $t \in \mathbb{R}$ with $\sqrt[4]{3.1} \approx 1.32$. Essentially, $\sqrt{17/12} > 1$ appears for asymmetric variables, since $\|\cdot\|_G$ is defined by comparing a Gaussian variable $G$ that is symmetric. A technical reason for this improvement is that $\|\cdot\|_G$ does not need Stirling's approximation for attaining a sharper MGF bound when expanding the exponential function by Taylor's formula. To show the tightness of Theorem 2(b), in Figure 5 of Appendix C, we gives some comparisons with $\sigma_{opt}(X)$, $\sqrt{17/12}\|X\|_G$, $\sqrt{2e}\|X\|_{\psi_2}$, $\|X\|_{w_2}/\sqrt{2}$ and $\sqrt{\mathrm{Var}\,X}$ in terms of confidence length in Table 1, when $X$ is Bernoulli or beta distribution.

## 4 ESTIMATION OF THE INTRINSIC MOMENT NORM

A first thought to estimate $\|X\|_G$ is by the plug-in approach. Although $k_X$ is proven to be finite in Lemma 2, its (possibly large) exact value is still unknown in practice. Instead, we use a non-decreasing *index sequence* $\{\kappa_n\}$ to replace $k_X$ in the estimation. Hence, we suggest a plug-in feasible estimator

$$\widehat{\|X\|}_G = \max_{1\le k\le\kappa_n}\left[\frac{1}{(2k-1)!!}\frac{1}{n}\sum_{i=1}^n X_i^{2k}\right]^{1/(2k)}. \tag{6}$$

Deriving the non-asymptotic property of the $\widehat{\|X\|}_G$ is not an easy task: the maximum point $\hat{k}(\kappa_n) := \arg\max_{1\le k\le\kappa_n}\left[\frac{1}{(2k-1)!!}\frac{1}{n}\sum_{i=1}^n X_i^{2k}\right]^{1/(2k)}$ will change with the sample size $n$ even $\kappa_n$ is fixed.

To resolve this, we first examine the oracle estimator defined as $\widetilde{\|X\|}_G = \left[\frac{1}{(2k_X-1)!!}\frac{1}{n}\sum_{i=1}^n X_i^{2k_X}\right]^{1/2k_X}$. Here, based on Orlicz norm $\|Y\|_{\psi_\theta} := \inf\{t > 0 : \mathrm{E}\exp\{|Y|^\theta/t^\theta\} \le 2\}$ of sub-Weibull r.v. $Y$ with $\theta > 0$ (Hao et al., 2019; Zhang & Wei, 2022), we present the non-asymptotic concentration of $\widetilde{\|X\|}_G$ around it ture value $\|X\|_G$.

**Proposition 1.** *Suppose $\{X_i\}_{i=1}^n \overset{i.i.d.}{\sim} X$ and $X$ satisfies $\|X\|_{\psi_{1/k_X}} < \infty$, then for any $t > 0$,*

$$\mathrm{P}\left(\left|\widetilde{\|X\|}_G^{2k_X} - \|X\|_G^{2k_X}\right| \le 2e\|X\|_{\psi_{1/k_X}}C(k_X^{-1})\left\{\sqrt{\frac{t}{n}} + \gamma^{2k_X}A(k_X^{-1})\frac{t^{k_X}}{n}\right\}\right) \ge 1 - 2e^{-t},$$

*where the constant $\gamma \approx 1.78$, and the constant functions $C(\cdot)$ and $A(\cdot)$ are defined in Appendix C.*

The exponential-moment condition $\|X\|_{\psi_{1/k_X}} < \infty$ is too strong for the error bound of $\widetilde{\|X\|}_G^{2k_X} - \|X\|_G^{2k_X}$ in Proposition 1, although it has exponential decay probability $1 - 2\exp(-t)$.

Except for the direct plug-in estimator, here we resort to the median-of-means (MOM, Page244 in Nemirovskij & Yudin (1983)) as the robust plug-in estimator of intrinsic moment norm. Let $m$ and $b$ be a positive integer such that $n = mb$ and let $B_1, \ldots, B_b$ be a partition of $[n]$ into blocks of equal cardinality $m$. For any $s \in [b]$, let $\mathrm{P}_m^{B_s} X = m^{-1}\sum_{i\in B_s} X_i$ for independent data $\{X_i\}_{i=1}^n$. The MOM version intrinsic moment norm estimator is defined as

$$\widehat{\|X\|}_{b,G} := \max_{1\le k\le\kappa_n}\mathrm{med}_{s\in[b]}\left\{\left[[(2k-1)!!]^{-1}\mathrm{P}_m^{B_s} X^{2k}\right]^{1/(2k)}\right\}. \tag{7}$$

As stated in Proposition 1, the naive plug-in estimator $\widehat{\|X\|}_G = \widehat{\|X\|}_{1,G}$ is not robust. MOM estimators (7) with $b \gg 1$ have two merits: (a) it only needs finite moment conditions, but the exponential concentration bounds are still

achieved; (b) it permits some outliers in the data. Non-asymptotic inferences require to bound for $\|X\|_G$ exactly by a feasible estimator $\widehat{\|X\|}_{b,G}$ up to a sharp constants. Next, we establish a high-probability upper bound for the estimated norm, if the data has $O \cup I$ outlier assumptions as follows.

- (M.1) Suppose that the data $\{X_i\}_{i=1}^n$ contains $n - n_o$ inliers $\{X_i\}_{i \in I}$ drawn i.i.d. according to a target distribution, and there are no distributional assumptions on $n_o$ outliers $\{X_i\}_{i \in O}$;

- (M.2) $b = b_O + b_S$, where $b_O$ is the number of blocks *containing at least one outliers* and $b_S$ is the number of *sane blocks containing no outliers*. Let $\varepsilon := n_o/n$ be the fraction of the outliers and $\frac{n_o}{b} < \frac{1}{2}$. Assume here exists a fraction function $\eta(\varepsilon)$ for sane block such that $b_S \geq \eta(\varepsilon)b$ for a function $\eta(\varepsilon) \in (0, 1]$.

To serve for error bounds in the presence of outliers, (M.2) considers the specific fraction function of the polluted inputs; see Laforgue et al. (2021). Define $\underline{g}_{k,m}(\sigma_k)$ and $\bar{g}_{k,m}(\sigma_k)$ as the sequences for any $m \in \mathbb{N}$ and $1 \leq k \leq \kappa_n$:

$$\bar{g}_{k,m}(\sigma_k) := 1 - \left[ \mathbb{E}X^{2k}/(2k-1)!! \right]^{-1/(2k)} \max_{1 \leq k \leq \kappa_n} \left[ -2[m/\eta(\varepsilon)]^{-1/2}\sigma_k^k/(\mathbb{E}X^{2k}) + \mathbb{E}X^{2k}/(2k-1)!! \right]^{1/(2k)}; \quad (8)$$

and $\underline{g}_{k,m}(\sigma_k) := [2[m/\eta(\varepsilon)]^{-1/2}\sigma_k^k/(\mathbb{E}X^{2k}) + 1]^{1/(2k)} - 1$. We obtain a robust and non-asymptotic CI for $\|X\|_G$.

**Theorem 3** (Finite sample guaranteed coverage). *Suppose* $\sqrt{\mathrm{Var}X^{2k}} \leq \sigma_k^k$ *for a sequence* $\{\sigma_k\}_{k=1}^{\kappa_n}$, *we have*

$$\mathrm{P}\left\{ \|X\|_G \leq [1 - \max_{1 \leq k \leq \kappa_n} \bar{g}_{k,m}(\sigma_k)]^{-1}\widehat{\|X\|}_{b,G} \right\} > 1 - \kappa_n \cdot e^{-2b\eta(\varepsilon)(1 - \frac{3}{4\eta(\varepsilon)})^2};$$

*and* $\mathrm{P}\{\|X\|_G \geq [1 + \max_{1 \leq k \leq \kappa_n} \underline{g}_{k,m}(\sigma_k)]^{-1}\widehat{\|X\|}_{b,G}\} > 1 - \kappa_n \cdot e^{-2b\eta(\varepsilon)(1 - \frac{3}{4\eta(\varepsilon)})^2}$ *for* $\kappa_n \geq \kappa_X$ *under (M.1-M.2)*.

Theorem 3 ensures the concentration of the estimator $\widehat{\|X\|}_{b,G}$ when $\kappa_n \geq k_X$ under enough sample. If $\eta(\varepsilon) = 1$ with $\varepsilon = 0$, then the data are i.i.d., which have no outlier, and outlier assumptions in M.1-M.2 can be dropped in Theorem 3. When the data is i.i.d. Gaussian vector, Proposition 4.1 in Auer et al. (2002) also gave a high-probability estimated upper bound for $\ell_p$-norm of the vector of Gaussian standard deviations, our result is for intrinsic moment norm.

In practice, the block number $b$ can be taken by the adaptation method based on the Lepski method (Depersin & Lecué, 2022). To guarantee high probability events in Theorem 3, it is required that the index sequence $\kappa_n$ should not be very large for fixed $b$. The larger $\kappa_n$ needs larger $b$ in blocks $B_1, \ldots, B_b$. In the simulation, we will see that an increasing index sequence $\kappa_n$ with slow rate will lead a good performance.

Finally, we compare our two estimators (6) and (7), as well as the estimator (4) in Figure 3. We consider the standard Gaussian and Rademacher variable distributed $X$, in the two case we have $\|X\|_G^2 = \sigma_{opt}^2(X) = \mathrm{Var}(X) = 1$. The following figure shows the performance of three estimators under sample $n = 10$ to $1000$ with $\kappa_n$ just chosen as $\lceil \log n \rceil$. For the MOM method, we use five blocks in this simple setting. For a more complex case, one can use Lepski's method to choose $b$ (see Page & Grünewälder (2021)), but some considerable computation cost may be introduced. From Figure 3, we know that the performance of the MOM estimator is best, while the naive estimator (4) is worst. For the high-quality data of extremely small sample size, we can apply the leave-one-out Hodges-Lehmann method (Rousseeuw & Verboven, 2002) for further numerical improvement; see Appendix B for details.

## 5 APPLICATION IN MULTI-ARMED BANDIT PROBLEM

In the multi-armed bandit problem (MAB), a player chooses between $K$ different slot machines (an $K$-armed bandit), each with a different unknown random reward r.v.s $\{Y_k\}_{k=1}^K \subseteq \mathbb{R}$, while each realization of a fixed arm $k$ is independent and shares the same distribution. Further, we assume the rewards are sub-Gaussian, i.e.

$$\|Y_k - \mu_k\|_G < \infty, \qquad k \in [K]. \quad (9)$$

Our goal is to find the best arm with the largest expected reward, say $Y_{t*}$, by pulling arms. In each round $t \in [T]$, the player pulls an arm (an action) $A_t \in [K]$. Conditioning on $\{A_t = k\}$, we define the observed reward $\{Y_{k,t}\}_{t \in [T]} \overset{\text{i.i.d.}}{\sim} P_k$. The goal of the exploration in MAB is to minimize the cumulative regret after $T$ steps:

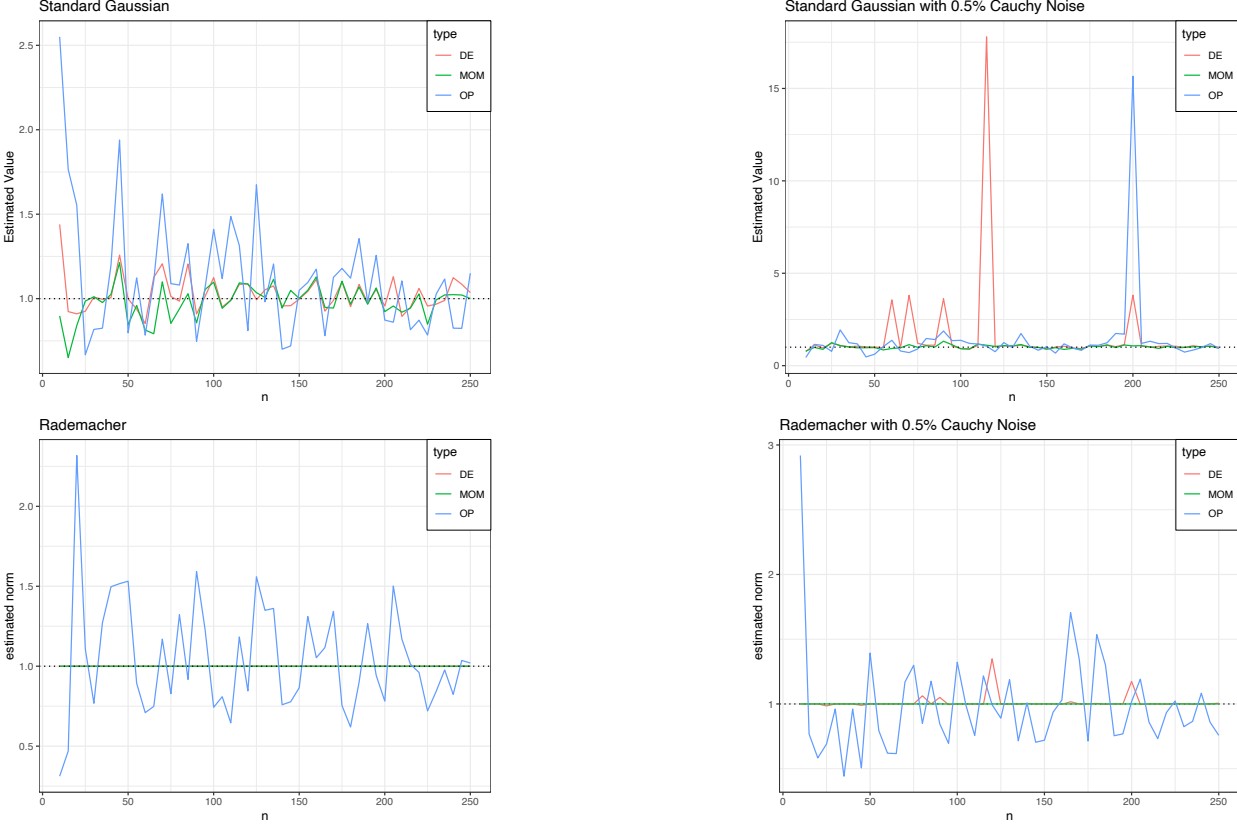

Figure 3: DE represents the naive plug-in estimator (6), MOM represents the MOM estimator (7), and OP is the estimator plug-in naive estimator (4) for the optimal variance proxy.

$\text{Reg}_T(Y, A) := \sum_{t=1}^{T}(\mu_{t^\star} - \mu_{A_t})$, i.e. the performance of any exploration strategy $\{A_t\}_{t\in[T]}$. The exploration performance is better, if we have smaller $\text{Reg}_T(Y, A)$. Without loss of generality, we assume $t^\star = 1$. We seek to evaluate the expected bounds from the decomposition (see Lemma 4.5 in Lattimore & Szepesvári (2020)),

$$\text{Reg}_T := \text{E}\,\text{Reg}_T(Y, A) = \sum_{k=1}^{K}\Delta_k\text{E}\left[\sum_{t=1}^{T}1\left\{A_t = k\right\}\right], \tag{10}$$

where E is taken on the randomness of the player's actions $\{A_t\}_{t\in[T]}$, and $\Delta_k = \mu_1 - \mu_k$ is the sub-optimality gap for arm $k \in [K]/\{1\}$. The upper bound of $\text{Reg}_T$ is called problem-independent if the regret bound depends on the distribution of the data and does not rely on the gap $\Delta_k$.

For each iteration $t$, let $T_k(t) := \text{card}\{1 \leq \tau \leq t : A_\tau = k\}$ be the number of pull for arm $k$ until time $t$ during the bandit process. Then if we define $\overline{Y}_{T_k(t)} := \frac{1}{T_k(t)}\sum_{\tau \leq t, A_\tau = k} Y_{k,\tau}$ as the running average of the rewards of arm $k$ at time $t$. Suppose we obtain a $100(1 - \delta)\%$ CI $\left[\overline{Y}_{T_k(t)} - c_k(t), \overline{Y}_{T_k(t)} + c_k(t)\right]$ for $\mu_k$ from a tight concentration inequality. Therefore, we confidently reckon that the reward of arm $k$ is $\overline{Y}_{T_k(t)} + c_k(t)$, and play the arm $A_t = k$, hoping to maximize the reward with a high probability for finite $t$. This is upper confidence bound (UCB, Auer et al. (2002)) algorithms. And many works based on this methods appears recently, for example, Hao et al. (2019) use bootstrap method with the second order correction to give a algorithm with the explicit regret bounds for sub-Gaussian rewards. However, many existent algorithms contain unknown norms for the random rewards, they are actually infeasible. And Theorem 4 is one example with explicit regret bound. For instance, the algorithm Hao et al. (2019) needs to use the unknown Orlicz-norm of $\overline{Y}_k - \mu_k$ in the algorithm. Thus, it is actually infeasible in practice.

Fortunately, our estimator can solve this problem. Suppose that $Y_k - \mu_k$ is symmetric around zero, by one-side version of Theorem 2, the (9) implies that for all $k$ and all $t$, $\text{P}(\overline{Y}_{T_k(t)} > \mu_k + \|Y_k - \mu_k\|_G\sqrt{\frac{2}{T_k(t)}\log\frac{1}{\delta}}) \leq \delta$. Let subsample size $m_k$ and block size $b_k$ be positive integer such that $T_k(t) = m_kb_k$ for MOM estimators $\widehat{\|Y_k - \mu_k\|}_{b_k, G}$ in

Section 3. Theorem 3 (a) guarantee that true norms can be replaced by MOM-estimated norms such that $P(\overline{Y}_{T_k(t)} \leq \mu_k + \frac{\|\widehat{Y_k - \mu_k}\|_{b_k,G}}{1-o(1)} \sqrt{\frac{2}{T_k(t)} \log \frac{1}{\delta}}) \geq 1 - \delta - k_{Y_k} \cdot \exp(-b_k/8)$ if $\eta(\varepsilon) = 1$ with $\varepsilon = 0$.

If the UCB algorithm is correctly applied, for a finite $T_k(t)$, with high probability, we will pull the best arm.

In practice, we nearly do not know any knowledge about the data. As a flexible way of uncertainty qualification, the multiplier bootstrap (Arlot et al., 2010) enables mimicking the non-asymptotic properties of the target statistic by reweighing its summands of the centralized empirical mean. The multiplier bootstrapped quantile for the i.i.d. observation $\mathbf{Y}_n := \{Y_i\}_{i=1}^n$ is the $(1-\alpha)$-quantile of the distribution of $n^{-1}\sum_{i=1}^n w_i(Y_i - \overline{Y}_n)$, which is defined as

$$q_\alpha(\mathbf{Y}_n - \overline{Y}_n, \mathbf{w}) := \inf\{x \in \mathbb{R} \mid P_w(n^{-1}\textstyle\sum_{i=1}^n w_i(Y_i - \overline{Y}_n) > x) \leq \alpha\},$$

where $\mathbf{w} := \{w_i\}_{i=1}^n$ are bootstrap random weights independent of $\mathbf{Y}_n$. We denote the statistics $\widehat{\varphi}_G(\mathbf{Y}_n)$ as something satisfying $P_{\mathbf{Y}_n}(|\overline{Y}_n - EY_1| \geq \widehat{\varphi}_G(\mathbf{Y}_n)) \leq \alpha$.

---

**Algorithm 1:** Bootstrapped UCB

---

Input: $\widehat{\varphi}_G(\mathbf{Y}_{T_k(t)})$ is given by (11).

    **for** $t = 1, \ldots, K$ **do**
      Pull each arm once to initialize the algorithm.
    **end**
    **for** $t = K+1, \ldots, T$ **do**
      Set a confidence level $\alpha \in (0, 1)$.
      Calculate the boostrapped quantile $q_{\alpha/2}(\mathbf{Y}_{T_k(t)} - \overline{Y}_{T_k(t)}, \mathbf{w})$ with the Rademacher bootstrapped weights $\mathbf{w}$ independent with any $Y$.
      Pull the arm
$$A_t = \underset{k\in[K]}{\arg\max}\, \text{UCB}_k(t) := \underset{k\in[K]}{\arg\max}\, \big(\overline{Y}_{T_k(t)} + q_{\alpha/2}(\mathbf{Y}_{T_k(t)} - \overline{Y}_{T_k(t)}, \mathbf{w}) + \sqrt{\tfrac{2\log(4/\alpha)}{T_k(t)}}\widehat{\varphi}_G(\mathbf{Y}_{T_k(t)})\big).$$
      Receive reward $Y_{A_t}$.
    **end**

---

Motivated by Hao et al. (2019), we design Algorithm 1 based on some estimators of the UCB. It guarantees a relatively small regret by bootstrapped threshold $q_{\alpha/2}(\mathbf{Y}_{T_k(t)} - \overline{Y}_{T_k(t)}, \mathbf{w})$ adding a concentration based second-order correction $\widehat{\varphi}_G(\mathbf{Y}_{T_k(t)})$ that is specified in Theorem 3. In the following regret bounds, we assume the mean reward from the $k$-th arm $\mu_k$ is known. In practice, it can be replaced by a robust estimator, and we obtain the results of MOM estimator.

**Theorem 4.** *Consider a $K$-armed sub-G bandit under (9) and suppose that $Y_k - \mu_k$ is symmetric around zero. For any round $T$, according to moment conditions in Theorem 3, choosing $\widehat{\varphi}_G(\mathbf{Y}_{T_k(t)})$ as*

$$\widehat{\varphi}_G(\mathbf{Y}_{T_k(t)}) = \frac{\sqrt{2\log(4/\alpha)}}{T_k^{1/2}(t) - 1}\|\widehat{Y_k - \mu_k}\|_{b_k,G} \tag{11}$$

*as a re-scaled version of MOM estimator $\|\widehat{Y_k - \mu_k}\|_{b_k,G}$ with block number $b_k$ satisfying the moment assumptions C[UCB1] and C[UCB2] in Appendix C. Fix a confidence level $\alpha = 4/T^2$, if the player pull an arm $A_t \in [K]$ according to Algorithm 1, then we have the problem-dependent regret of Algorithm 1 is bounded by*

$$\text{Reg}_T \leq 16(2 + \sqrt{2})^2 \max_{k\in[K]} \|Y_k - \mu_k\|_G^2 \log T \sum_{k=2}^K \Delta_k^{-1} + (4T^{-1} + 2T^{-25-16\sqrt{2}} + 8)\sum_{k=2}^K \Delta_k,$$

*where $\Delta_k$ is the sub-optimality gap. Moreover, let $\mu_1^* := \max_{k_1\in[K]} \mu_{k_1} - \min_{k_2\in[K]} \mu_{k_2}$ be the range over the rewards, the problem-independent regret*

$$\text{Reg}_T \leq 8(2 + \sqrt{2}) \max_{k\in[K]} \|Y_k - \mu_k\|_G \sqrt{TK\log T} + (4T^{-1} + 2T^{-25-16\sqrt{2}} + 8)K\mu_1^*.$$

From Theorem 4, we know that the regret of our method achieve minimax rate $\log T$ for a problem-dependent problem and $\sqrt{KT}$ for a problem-independent case (see Tao et al. (2022)), so Algorithm 1 can be seen as an optimal algorithm.

Compared with the traditional vanilla UCB, we do improve the constant. When $Y_k \sim N(\mu_k, 1)$, the constant factor in regret bound in Auer et al. (2002) is 256, which is larger than $16(2 + \sqrt{2})^2$ in our theorem.

When the UCB has unknown sub-G parameters, Theorem 4 first studies a feasible UCB algorithm with sub-G parameter plugging estimation. Many previous UCB algorithms based on non-asymptotic inference in the literature assume that the sub-G parameter is a preset constant, see the algorithm in Hao et al. (2019) for instance.

Next, we give an simulation for Theorem 4 in two sub-G cases to verify the performance of estimated norms. Similar to Hao et al. (2019); Wang et al. (2020), we design the three methods as follows:

1. Use our method $\widehat{\varphi}(\mathbf{Y}_{T_k(t)})$ with *Estimated Norm* in Theorem 4;

2. Use *Asymptotic Naive varphi* $\widetilde{\varphi}(\mathbf{Y}_{T_k(t)})$ satisfying $P(|\overline{Y}_{T_k(t)} - \mu_k| \leq \widetilde{\varphi}(\mathbf{Y}_{T_k(t)})) \to \alpha$ by CLT, i.e. $\widetilde{\varphi}(\mathbf{Y}_{T_k(t)}) = \widehat{\sigma}_k \Phi^{-1}(1 - \alpha/2)/\sqrt{T_k(t)}$ with $\widehat{\sigma}_k = \sqrt{\frac{1}{T_k(t)} \sum_{\tau \leq t, A_\tau = k} (Y_{k,\tau} - \overline{Y}_{T_k(t)})^2}$ as the estimated standard deviation;

3. Regard all the unbounded rewards as bounded r.v. and use Hoeffding's inequality (*wrongly use Hoeffding's inequality*) to construct $\varphi$, i.e. $\check{\varphi}(\mathbf{Y}_{T_k(t)}) = \left[ \max\{\mathbf{Y}_{T_k(t)}\} - \min\{\mathbf{Y}_{T_k(t)}\} \right] \sqrt{\frac{\log(2/\alpha)}{2T_k(t)}}$.

And for our detailed MAB simulation, we consider as follows, in each case, the number of arms is assigned as $K = 5$, and $(\mu_1, \ldots, \mu_5) = (0.1, 0.05, 0.02, 0.01, 0.01)^\top$,

$$\text{EG1. } Y_k \sim N(\mu_k, \mu_k^2); \qquad \text{EG2. } Y_k \sim 0.9N(\mu_k, \mu_k) + 0.1N(\mu_{k+1}, \mu_{k+1}) \text{ where } \mu_6 := \mu_5.$$

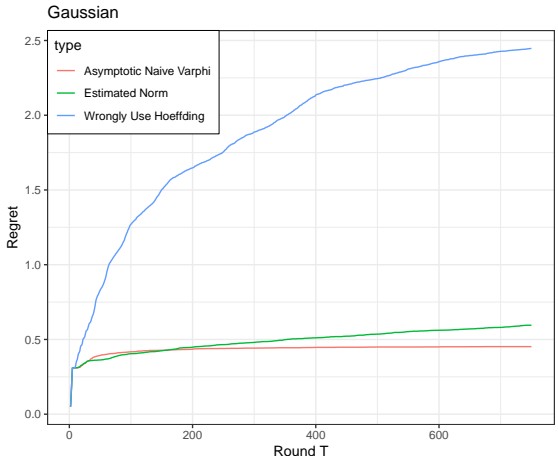
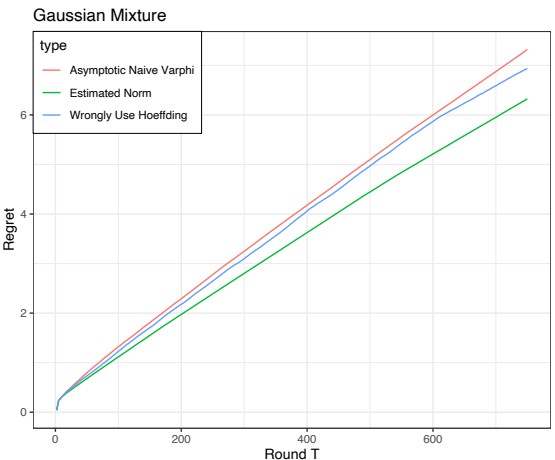

Figure 4: The regret of MAB with sub-G rewards under three methods. $x$-axis represents the round and $y$-axis is the cumulative regret.

As we can see, EG1 and EG2 are both sub-Gaussian rewards. In the simulation, $\mu_k$ is assigned sightly small for bounded $\max_k \Delta_k$, which is a standard-setting in the MAB problem (see the condition of Corollary 1 in Wang et al. (2020) for instance). The simulation results are shown in Figure 4, which illustrates that our method outperforms the other two methods under the unbounded sub-Gaussian rewards and small sample $T \in [1, 150]$. In the Gaussian case, Algorithm 1 can also give better results compared with the CLT-based UCB when the round is relatively small. For the Gaussian mixture case, the algorithm based on the intrinsic moment norm has smaller regret than the other two methods under $T \in [1, 800]$.

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
