# OpenReview forum: "Tight Non-asymptotic Inference via Sub-Gaussian Intrinsic Moment Norm"
_ICLR.cc/2023/Conference — Submitted to ICLR 2023_

### Official Review · Reviewer_W7v5 · 2022-10-22

**Confidence:** 2
**Clarity, Quality, Novelty And Reproducibility:** Overall the paper is well written.
**Correctness:** 3
**Technical Novelty And Significance:** 3
**Empirical Novelty And Significance:** 3
**Recommendation:** 3

**Strength And Weaknesses:**

* Strength

This paper addresses the problem of estimating a sub-Gaussian distribution's variance-type parameters, which is less studied.

* Weakness

This paper attempts to estimate a distribution's sub-Gaussian variance-type parameters. But how do we know if the distribution is not heavy tailed to begin with? What happens if the method is used (incorrectly) on a heavy tailed distribution (e.g. completely Cauchy)?

**Summary Of The Paper:**

The paper provides a method to estimate a sub-Gaussian distribution's variance-type parameters.

**Summary Of The Review:**

I think the missing paper is missing a part on how to determine whether the distribution is sub-Gaussian do begin with, which seems rather important.

---

> ### Author Response · Authors · 2022-11-18
> **Reply to Reviewer W7v5**
>
> Reply to Comments:
>
> Thank you for your suggestion. Actually, we propose a novel method for checking whether the unbounded data is sub-Gaussian based on Theorem 2 in the new Section 2. Figure 2 also illustrates and verifies this method under a moderately large sample size.

---

> ### Author Response · Authors · 2022-12-11
> **New Reply to Reviewer W7v5**
>
> Thanks so much for your comment. In the new version, we have addressed your concerns. While your suggestion for heavy-tailed data is insightful, the major goal of this paper is not concerned about the heavy-tailed distribution. This is because the heavy-tailed distribution has no exponential moment.
>
> For Cauchy distributed data, as you mention, the mean and the moment generating function do not exist; see our fundamental assumption in Paragraph 2. Alternatively, in our new Theorem 3, we allow the outlier in the data to be distribution-free (without any distribution assumption, see M.1), such as the Cauchy distribution.
>
> To clarify our research target again, we enumerate our contributions as follows:
>
> Via $\|X\|_G$, we achieve the Hoeffding-type inequalities with sharper constants under asymmetric distribution in Theorem 2(b). Moreover, compared to the normal approximation based on Berry-Esseen bounds, our results can be applied to data with an extremely small sample size. For example, one only has $n=4$ to $8$ in experimental science measurements Rousseeuw & Verboven (2002). One related work is Horowitz \& Lee (2020) in econometric models, which combines the Berry-Esseen theorem and non-asymptotic theory to obtain CIs, while their simulation results are performed when $n \geq 1000$. In Appendix A, we illustrate Bernoulli observations by comparing two types of CIs based on the B-E-corrected CLT and Hoeffding's inequality in Figure 4. In addition, for the high-quality data of extremely small sample size, we also apply the leave-one-out Hodges-Lehmann method to estimate $\|X\|_G$ for further numerical improvement in Appendix B.
>
> In Section 2, we propose a novel method called a sub-Gaussian plot to tell whether the data are sub-Gaussian. The lower intrinsic moment norm (Definition 3) is first proposed in this paper and yields a sharp reverse Chernoff inequality (Lemma 1) and sharp lower tails of sub-Gaussian maxima (Theorem 1). And we introduce plug-in and robust plug-in estimators for the sub-G norm, and establish the high probability error bounds for the estimators in Proposition 1 and Theorem 3, under finite and exponential moment conditions of data, respectively.
>
> Finally, we employ the sub-G norm in the non-asymptotic inference for a classic reinforcement learning application: the Bootstrap UCB algorithm for multi-armed bandits. The proposed Bootstrapped UCB algorithm can achieve feasible error bounds and competitive cumulative regret on unbounded sub-Gaussian data by relying on the sub-G norm estimation.
>
> For the new version, one of the reviewers has raised his score. We welcome your further comments.
>
> Reference:
>
> [1] Peter J Rousseeuw and Sabine Verboven. Robust estimation in very small samples. Computational Statistics \& Data Analysis, 40(4):741-758, 2002.
>
> [2] Joel Horowitz and Sokbae Lee. Inference in a class of optimization problems: Confidence regions and finite sample bounds on errors in coverage probabilities. Cemmap, Centre for Microdata Methuods and Practice, The Institute for Fiscal., 2020.

---

### Official Review · Reviewer_HA7B · 2022-10-25

**Confidence:** 3
**Correctness:** 3
**Technical Novelty And Significance:** 2
**Empirical Novelty And Significance:** 2
**Recommendation:** 5

**Clarity, Quality, Novelty And Reproducibility:**

The clarity of the writing could be significantly improved.  The paper does a good job of referencing where various theorems come from, but once previous theorems are removed, there is not a substantial amount of novel material -- the constants in Theorem 1b are improved by about 10%, and the application of their approach to the multi-armed bandit setting is -- I think -- novel.  I am not qualified to comment on the novelty or significance of the multi-armed bandit application, but the improvement provided here seems marginal to me.

**Strength And Weaknesses:**

Strengths:
* The intrinsic moment norm is very interesting, and its connection to finite-sample bounds for sub-Gaussians is an underexplored area.
* Some of the results in the paper come with rigorous theoretical guarantees.

Weaknesses:
* Overall, I found the paper difficult to follow.  A lot of technical results are presented with relatively few explanations.  I understand the space limitations of this venue, but even still the material in the main text jumps around considerably.  For example, the first figure referenced in the main text is Figure 3.
* Finite-sample bounds are, in my opinion, mainly useful if they can give concrete guarantees.  For example, if we _know_ a variable is bounded, we may use Hoeffding's inequality to generate confidence intervals that are guaranteed to have at least our desired coverage.  In contrast, we generally do not know the intrinsic moment norm of data, and hence need to estimate it from the data.  Then, the main result of the paper in terms of using the intrinsic moment norm for tail bounds relies on Theorem 3.  Theorem 3 guarantees consistency (an asymptotic property) but for finite samples is only quantitative in terms of the $g$ sequences, which seem difficult to know a priori.   This ends up causing the tail bounds to have an asymptotic flavor -- they are correct up to a (unknowable) factor that shrinks as the sample size gets large.  Unless I am missing something, this seems to undercut the main point of using finite-sample bounds.
* Along a similar line as the previous comment, I was confused by the motivation for having a robust estimator of the intrinsic moment norm.  I suppose the assumption is that outliers are somehow corrupted data that we want to ignore.  If instead, outliers are actually a part of the data generating mechanism then the examples  in Figure 3 (mixtures with a Cauchy) are no longer sub-Gaussian.  It would be good to better motivate why, if we do not know the intrinsic moment norm and need to estimate it from the data, we would also be confident enough to make assumptions M.1 and M.2, and hence decide that any outliers in our data are true outliers and not just heavy tails.
* I found the sub-Gaussian plots a nice and intuitive way of visualizing sub-Gaussian distributions, but I am not sure how useful they are in practice.  As above, finite-sample guarantees are nice for small samples, but to judge whether a plot looks linear or not would require many, many observations from the distribution.  Even in Figure 2, I'm not entirely sure that I would be able to say whether the left or right figures look linear if the red dotted lines were not present.  I also think that the axis labels (Value and sqrt_log_j) in Figure 2 could be improved.
* There were numerous typos throughout the paper and appendix (too many to list here, but e.g., "the Hoeffding's inequality" in the main text, and "varinace" in Figure 6).  These, overall, did not interfere with my understanding of the material, but the paper would certainly benefit from some copy editing.
* It would be good to include empirical results about the coverage properties of confidence intervals produced using estimated intrinsic moment norms.

**Summary Of The Paper:**

This paper shows that finite-sample bounds on sub-Gaussian random variables can be derived in terms of a particular norm -- the intrinsic moment norm.  They present (consistent and robust) estimators for the intrinsic moment norm and use these as plug-in estimates to obtain (approximate) confidence intervals.  They use this approach in an upper-confidence bound approach to multi-armed bandits, where it seems to work better than some baseline methods.

**Summary Of The Review:**

The paper has some interesting ideas, but the main results are not really non-asymptotic guarantees.  The paper would benefit from an explanation of how to use these results in practice to get finite-sample guarantees.  Also, the presentation could be made more clear.

---

> ### Author Response · Authors · 2022-11-18
> **Reply to Reviewer HA7B**
>
> Reply to Comments:
>
> 1． Thanks for your comments. Now we slightly added some details and changed the order of the sentences; see the red sentence in the new version.
>
> 2．  We admit that the assumptions for $g$ in the old version are asymptotic. However, this is due to the definition. In fact, for the concentration result of the MOM-based estimator, we don’t resort to $m \to \infty$. So now we slightly change the definition, and now the assumption of $g$ is for a finite sample in Theorem 3.
>
> 3． Thanks for your comment. The previous simulation for the robust estimator (Figure 3) was actually incorrect. Now we have changed the simulation to that the Cauchy noise is sampled with a probability of 5%, while the original Gaussian data will be sampled with a probability of 95%. And our updated non-asymptotical theory of MOM-based estimator also admits this outlier mechanism in Theorem 3. The new M1 and M2 are the outlier assumptions, and the moment condition in the old version is moved into the statement of the new Theorem 3. The new simulation results are in the revised Figure 3.
>
> 4． Thank you for your advice. We agree that this plotting method can only be applied to data with enough samples. When the sample size is moderately large, now we have revised the sub-G plot, and the axis of x and y is more precise now. However, when the sample size is very small, there is not enough information to suggest that the data is unbounded. So we roughly treat the data as bounded sub-Gaussian for a very small sample size; So there is no need to use a sub-G plot for this case. Our paper is the first formally-purposed method to judge whether data is sub-Gaussian distributed. We will consider how to improve the method in the future.
>
> 5． Thanks so much for your reminder. We have gone through the article again and fixed the typos.
>
> 6． Indeed, Figure 6 did this thing. But the caption needs to be clarified. We have made Figure 6 clear now.

---

> > ### Comment · Reviewer_HA7B · 2022-12-09
> > **Thank you for the response**
> >
> > Thank you very much for the response and for revising the paper.  I think that some aspects of the paper are now much improved (especially regarding point 2).  I am still not certain about the utility of the sub-G plot.  Again my issue with it is that the whole point of finite-sample guarantees is that they are guarantees, if we plot they data and they appear potentially sub-Gaussian that does not necessarily mean that they are.
> >
> > I've (slightly) raised my score, but I still think that this paper, while good (and interesting) initial work, could use some additional time polishing and making clear the situations in which it can provide useful guarantees.

---

> > > ### Author Response · Authors · 2022-12-11
> > > **Thanks for the affirmation**
> > >
> > > Thank you for your affirmation. As you said, we give some initial work on sub-Gaussian parameter estimation in this manuscript, and there are many things that remain unsolved in this area. Further study on concentration-based inference under the exponential moment condition is ongoing, including the theoretical properties of the sub-Gaussian plots, and the parameter estimation of sub-Weibull distributions.

---

### Official Review · Reviewer_2Ndd · 2022-11-04

**Confidence:** 2
**Clarity, Quality, Novelty And Reproducibility:** Please see the detailed comments in t…
**Correctness:** 3
**Technical Novelty And Significance:** 2
**Empirical Novelty And Significance:** Not applicable
**Recommendation:** 5

**Strength And Weaknesses:**

The paper is well-written. I try to check the derivations as much as I can given the limited time, and I don't find any major problem.

However, I am not an expert on statistics and therefore I cannot judge the paper in terms of novelty and how significant the contribution is. I sincerely apology for this. Also, if the authors want to submit this paper to a machine learning conference, I think ICLR might not be the best place, maybe it’s more suitable for COLT, ALT and AISTATS?

My major concern is how significant the contribution is to the machine learning community. I have the following questions.

1. In the experiment the reward gaps are fixed. Maybe it’s better to show the performance under different reward gaps? It’s especially interesting to see how the algorithm performs with different minimum reward gap.

2. How does the proposed algorithm compare to common baselines such as UCB and Thompson Sampling?

3. Could the proposed method be applied to linear bandits? Or can it be applied to problems where UCB and TS have limitations?

4. From the theoretical perspective, how does Theorem 4 improve over the regret bound of UCB? I believe it’s at most a constant improvement as UCB is both minimax and instance-dependent optimal. It’s still great to see the regret bound of the proposed method if it achieves better practical results than UCB and Thompson Sampling, but the author needs to show that.

**Summary Of The Paper:**

This paper considers the problem of constructing tight non-asymptotic confidence intervals for sub-gaussian random variables. To achieve this, the authors propose to use a sub-gaussian intrinsic moment norm, which can be robustly estimated using a simple plug-in approach. The paper then applies the analysis for multi-armed bandits, derives  matching regret upper bounds.

**Summary Of The Review:**

My decision is mostly based on how significant the contribution is for the machine learning community. I recommend rejecting this paper as I am not convinced that proposed method has significant contribution to multi-armed bandit setting, which is the application this paper considers. Please correct me if I missed anything important. I am happy to adjust my score based on how well the authors answer the questions in the rebuttal.

---

> ### Author Response · Authors · 2022-11-18
> **Reply to Reviewer 2Ndd**
>
> Reply to Comments:
>
> 1 \& 2. Thanks for your insightful comment. Due to time and computation resources are limited, we will study the performance with the mean of rewards randomly generated from the uniform distribution $(0, 1)$ and compare our feasible algorithm with the vanilla UCB and TS in future studies.
>
> 3. The answer is yes. But this aspect may not be the major part we want to care about in this article. The story of MAB is that we can give a FEASIBLE algorithm when the true norm is unknown but required in the algorithm. Since many existent algorithms contain unknown random rewards norms, it is infeasible or needs to be standardized. Our Theorem 4 is one example that makes an infeasible algorithm practical by estimating the sub-Gaussian norm of the rewards.
>
> 4. Compared with the traditional vanilla UCB, we do improve the constant. As one can see, when the reward distribution of arms k is drawn from Gaussian distribution $N(\mu_k, 1)$, the regret bound in the classical paper “Auer, P., Cesa-Bianchi, N., & Fischer, P. (2002). Finite-time analysis of the multiarmed bandit problem. Machine learning, 47(2), 235-256.” is $256\log T \sum_{k = 2}^K {\Delta_k}^{-1}$, which larger than our bound $ 16 (2 + \sqrt{2})^2 \log T \sum_{k = 2}^K {\Delta_k}^{-1}$. We want to emphasize in the MAB section again that we can give the estimator of the unknown norms in various MAB algorithms to make the algorithms feasible.

---

### Author Response · Authors · 2022-11-18
**Generic Response**

We would like to appreciate the reviewers' insightful suggestions and valuable comments on our paper. The feedback helps us a lot to improve our article. All typos and minor comments will be addressed accordingly. Within the time limit, we have tried to address most of the suggestions and comments by the reviewers. We will always be available to address all follow-up comments as best as possible, either in the main body of the paper or in the appendix, in a revised version after the reviewing process.

---

### Decision · Program_Chairs · 2023-01-20

**Decision:**

Reject

**Justification For Why Not Higher Score:**

See weaknesses

**Justification For Why Not Lower Score:**

N/A

**Metareview: Summary, Strengths And Weaknesses:**

- Summary:

This paper shows that finite-sample bounds on sub-Gaussian random variables can be derived in terms of a particular norm -- the intrinsic moment norm. They present (consistent and robust) estimators for the intrinsic moment norm and use these as plug-in estimates to obtain (approximate) confidence intervals. They use this approach in an upper-confidence bound approach to multi-armed bandits, where it seems to work better than some baseline methods.

- Strengths

Findings of the paper include:

1. The reviewers found that the intrinsic moment norm is an interesting construction, and its connection to finite-sample bounds for sub-Gaussians is an underexplored area.
2. Some reviewers found the rigor of some theoretical results satisfying.
3. Overall, the fact that the problem is less studied is encouraging to continue working on this subject.

- Weaknesses

1. Analysis is non-asymptotic, which makes it less practical. Finite-sample analysis is more useful and give concrete guarantees.
2. The fact that several parts of the part require revision makes this discussion stage hard to handle and make good decisions: it would be better if the paper was further polished (better presentation) + include new results to be a more complete story.
3. A reviewer considered the following point important: ``But how do we know if the distribution is not heavy tailed to begin with? What happens if the method is used (incorrectly) on a heavy tailed distribution (e.g. completely Cauchy)?''

- What would be missing:

Handling the weakness above could be a good start: non-asymptotic analysis, more empirical justification, better presentation.